# Effect of Deep Cryogenic Treatment on Microstructure and Wear Resistance of LC3530 Fe-Based Laser Cladding Coating

**DOI:** 10.3390/ma12152400

**Published:** 2019-07-27

**Authors:** Xiao Zhang, Yajun Zhou

**Affiliations:** 1College of Mechanical and Electrical Engineering, Central South University, Changsha 410083, China; 2National Key Laboratory of High Performance Complex Manufacturing, Central South University, Changsha 410083, China

**Keywords:** deep cryogenic treatment, laser cladding, grain refinement, wear resistance, wear morphology

## Abstract

The effect of deep cryogenic treatment on microstructure and wear resistance of LC3530 Fe-based powder laser cladding coating was investigated in this paper. The cladding coating was subjected to deep cryogenic treatment for the different holding times of 3, 6, 9, 12, and 24 h, followed by tempering at room temperature. Microstructure of the cladding coating was observed by optical microscope (OM) and the microhardness was measured by the Vickers-hardness tester. The wear was tested by ball and flat surface grinding testing conducted on the material surface comprehensive performance tester. The wear scars were analyzed using a non-contact optical surface profiler and scanning electron microscope (SEM). The results showed the grain size of cladding coating after 12 h of deep cryogenic treatment was significantly reduced by 36.50% compared to the non-cryogenically treated cladding coating, and the microhardness value increased by approximately 34%. According to the wear coefficient calculated by the Archard model, the wear resistance improved about five times and the wear mechanism was micro-ploughing. The deep cryogenic treatment could enhance the wear resistance of the cladding coating by forming a wear resistant alloy compound and higher surface microhardness.

## 1. Introduction

Laser cladding, which involves spraying/coating high hardness and wear-resistant coating on the surface of a material on the basis of ensuring good mechanical properties of the material itself, has been a research hotspot [1]. Laser cladding technology rapidly heats and melts alloy powder or ceramic powder and the substrate surface under the laser beam. After the laser beam is removed, the material can form a surface coating with a very low dilution rate and metallurgical bonding with the substrate material by self-excited cooling. Compared with traditional surface strengthening processes technology, laser surface strengthening is a new type of non-polluting and efficient surface modification technology. It concentrates energy and has a small heat-affected zone, and non-contact processing does not cause mechanical extrusion or mechanical stress on the material. Finally, it has good processing flexibility, is not limited by the size of the workpiece, and can be used as the final processing procedure of the materials and parts [2]. Therefore, laser cladding can produce high performance alloy surfaces on cheap metal substrates without affecting the properties of the substrate, and thereby reducing cost and saving precious rare metal materials. Overall, a surface strengthening method for significantly improving wear resistance and mechanical properties of the substrate surface has been obtained. Based on this, the world’s advanced industrial countries have attached great importance to the research and application of laser cladding technology. The range of fields where laser cladding technology can be applied is very wide, covering almost the entire machinery manufacturing industry, including coal, petroleum, electric, railway, automobile, aviation and other industries. Fu [3] investigated the influence of laser cladding Fe-based alloys on the microstructure and wear characteristic of wheel/rail materials using a rolling–sliding wear testing apparatus. The result showed it could enhance the hardness and wear resistance of wheel/rail materials and prolong their wear life. Yakovlev et al. [4] reported that the prepared coating had high hardness and a low coefficient of friction (average microhardness was 730 HV_0.3_, average friction coefficient was 0.12 at room temperature) by using Nd: YAG laser to coat a metal-based wear-resistant composite coating containing self-lubricating material (CuSn) and reinforcing phase ceramic material (WC/Co).

In addition, deep cryogenic treatment (DCT) technology has been widely used to enhance the mechanical properties of cutting tools, measuring tools, molds and precision parts. Yifeng Yao et al. [5] explored the effects of DCT on the mechanical properties and the microstructure of GB 35CrMoV steel, and found that the longer the cryogenic treatment time, the higher the microhardness and the lower the wear ratio. Leandro Brunholi Ramos et al. [6] analyzed the tribocorrosion and electrochemical behavior of DIN 1.4110 martensitic stainless steels after cryogenic heat treatment, and discovered the treated samples presented higher hardness and lower corrosion current density (i_corr_). In other studies, Masoud Sehri et al. [7] concluded that the wear resistance of deep cryogenic treated samples was significantly higher (about 25%), because the DCT removed the retained austenite and formed a uniform fine carbides distribution in the matrix.

From the above studies, it can be concluded that laser cladding and DCT can both enhance the wear resistance of the material [8,9,10]. Nevertheless, the effect of DCT on laser cladding coating has not been reported yet. More and more materials are reinforced with laser cladding, but the mechanical properties of the cladding still need to be improved for industrial needs because the uneven composition and high thermal stress generated during the laser cladding process tend to make the hardness and wear resistance of the cladding coating not meet the expected requirements. However, the improvement effect of DCT technology is the overall effect on the interior of the workpiece, not limited to the surface. Therefore, when the cladding coating is subjected to DCT, the existing modification effect of the workpiece is not affected and the shape of the workpiece is not changed, and the mechanical properties of the cladding coating can be improved. GB 35CrMo is a common alloy structural steel with high static strength, impact toughness and high fatigue strength. It is widely used in the manufacture of various important workpieces, such as gears, crankshafts and engine spindles, and the surfaces of these workpieces usually need to be strengthened with laser cladding. When the substrate is cast iron or medium-low carbon steel, the Fe-based alloy powder is often used for laser cladding, which can meet the wear resistance requirements of the workpiece, and the price of it is cheap. Hence, the effect of DCT on microstructure and wear resistance of LC3530 Fe-based powder laser cladding coating is investigated in this paper.

In this study, the LC3530 Fe-based powder was firstly cladded on the surface of GB 35CrMo steel, then all the cladding coatings were treated in liquid nitrogen for different durations (0 h, 3 h, 6 h, 9 h, 12 h and 24 h, respectively). The microstructure was observed under optical microscope (OM) and the wear resistance was investigated by reciprocating wear tests. The wear behavior of cladding coating was explored by means of a non-contact optical surface profiler and scanning electron microscope (SEM).

## 2. Materials and Methods

### 2.1. Materials

The metal substrate was from GB 35CrMo steel, which is used for the manufacture of gears and crank shafts. The laser cladding powder was LC3530 Fe-based powder, which forms a good metallurgical bond with the substrate, with a particle size distribution of 45–180 μm, supplied by Xiangtan City Unicom Industry and Trade Co., Ltd. (Xiangtan, China). Chemical compositions are given in Table 1 and Table 2, respectively.

### 2.2. Laser Cladding and Deep Cryogenic Treatment

Prior to laser cladding, the surface of the GB 35CrMo steel was ground and polished, and then cleaned with alcohol and then water to remove the pollution. After that, the LC3530 Fe-based powder was cladded on GB 35CrMo steel by a TFL-H600 transceiver flow CO_2_ laser device (Tongfa, Shenzhen, China) and, using argon gas during the laser cladding. Laser power was 1350 W, scanning speed was 10 mm/s with a 5 mm diameter laser spot and 50% overlap ratio.

After laser cladding, the samples were made into 15 mm × 15 mm × 18 mm samples by means of a wire cutting machine (DK7735, Jiangzhou CNC Machine Tool Manufacturing Co. LTD, Taizhou, China). Next, DCT was carried out in liquid nitrogen (LN2). The cryogenic temperature was −196 °C and soaking time was 3 h (designed as 3 HCT), 6 h (6 HCT), 9 h (9 HCT) 12 h (12 HCT), and 24 h (24 HCT). The samples were restored to room temperature in air. Moreover, the sample without DCT was used as a reference and was labeled as 0 HCT. The whole experiment process profile is illustrated in Figure 1.

### 2.3. Microstructure Observation

The samples surfaces were all etched for 60 s with aqua regia (HCl:HNO_3_ = 3:1) in order to observe the microstructure. Surfaces after corrosion required cleaning with alcohol and water, respectively, to remove the residual corrosion residue. The microstructure was observed by an Olympus DSX500 (OLYMPUS Corporation, Tokyo, Japan) optical microscope, and measured according to the metallographic standard.

### 2.4. Microhardness Test and Wear Test

The microhardness of surfaces at room temperature was measured by a model HV-1000A microhardness tester (HV-1000A, Huayin experimental instrument Co., Changsha, China) with a loading force of 5 N for 15 s. A total of 10 microhardness points (each point was 0.2 mm apart) were tested on each surface to get accurate measurements. The microhardness measurement position was located in the center of the surface, a total of two lines with five test point per line were taken. Because the center had the best cladding effect, it could reduce the impact of the laser cladding process on the measurement.

The effect of DCT on the wear resistance of cladding coating under dry conditions was analyzed by using a material surface comprehensive performance tester. Reciprocating friction mode was adopted as shown in Figure 2 and the wear parameters were as follows: friction force was 10 N, wear time was 5 min, reciprocating wear length was 5 mm and reciprocating motor speed was 300 r/min. The grinding material was the GCr15 steel ball of 6 mm diameter.

### 2.5. Wear Surface Profile and Wear Scar Morphology

The three-dimensional profile of the wear surface was characterized by a non-contact optical surface profiler (Wyko NT9100, Veeco Instruments Inc., New York, New York State, USA) at a magnification of 5×. The wear scars were analyzed by the scanning electron microscope (SEM, Phenom ProX, Phenom World, Eindhoven, The Netherlands) and energy dispersive spectrometer (EDS).

## 3. Results and Discussion

### 3.1. Microstructure

Figure 3 shows the microstructure of cladding coatings with and without DCT. For the microstructure without DCT, the dendrite and equiaxed crystal mixing zone was formed on the surface, as shown in Figure 3a. However, the dendrites were gradually homogenized as the DCT time increased [11,12,13] (Figure 3b–d). Among them, the microstructure homogenization effect after 6 and 9 h of DCT was the best. However, the microstructure turned into a dendrite after 12 h DCT and formed a dendrite and equiaxed crystal mixing zone again after 24 h DCT (Figure 3e,f). Furthermore, the samples all contained spherical carbides which showed that the DCT promoted the precipitation of carbides in the cladding coating.

The results of the average grain size are shown in Figure 4. The grain size was reduced by 22.22%, 38.75%, 35.30%, 36.50% and 30.38% after 3 h, 6 h, 9 h, 12 h, and 24 h of DCT, respectively. The results indicated that the DCT process obviously improved the microstructure of the cladding coating as the microstructure was not only more uniform but also more refined. This was because the temperature was drastically lowered during DCT, so that lattice contraction occurred and generated internal stress according to the characteristics of thermal expansion and contraction of most metals. However, when the duration time was short (≤3 h), the internal stress caused by the temperature difference was small, which was not enough to start the dislocation. When the duration time was long enough (≥3 h), the internal stress accumulated to a certain extent, and the grain was refined by the resulting dislocation [14]. Because of this, the DCT could make the microstructure of cladding coating finer and more homogeneous. Considering the grain size and standard deviation, 6 h, 9 h, or 12 h DCT was optimal for the cladding coating.

### 3.2. Microhardness

Microhardness is an important indicator for measuring the wear resistance of a material [15,16]. Generally speaking, the higher the microhardness value of the material, the better the wear resistance of the material [17,18,19,20]. Therefore, the surface microhardness was measured at room temperature. Test points were selected with a spacing of 1 mm between each point and a total of 10 points were counted. The average microhardness and the standard deviation are represented in Figure 5.

According to the results, the microhardness gradually increased with the DCT duration time extension, and the average microhardness of the 0 h, 3 h, 6 h, 9 h, 12 h, and 24 h samples were 518.91 HV_0.5_, 620.34 HV_0.5_, 667.12 HV_0.5_, 682.79 HV_0.5_, 692.64 HV_0.5_, and 693.99 HV_0.5_. The microhardness value after DCT increased by approximately 20% (3 HCT), 29% (6 HCT), 32% (9 HCT), 33% (12 HCT), and 34% (24 HCT). This indicated that the DCT not only increased the microhardness, but also improved the uniformity of the microhardness distribution (Figure 5). The increase in microhardness was not only due to the formation of a more uniform and finer grain structure after DCT, but also to the precipitation of fine carbides [21] in the cladding coating (Figure 3). The dispersion of microhardness was caused by the difference in the microstructure and carbide content at different positions. If the microstructure at this position was uniform and the carbide content was large, the microhardness was high. Furthermore, the microhardness increased significantly before 6 h of DCT, but the increase in microhardness after 6 h was slow. This was because the microhardness is determined by the microstructure and carbides. The change in grain size and carbides precipitation after 6 h was small, so the increment in microhardness value was slow. In summary, the surface microhardness continued to increase as the DCT duration time increased, but the increment became slower and smaller.

### 3.3. Coefficient of Friction

The wear tests under dry conditions were conducted in a material surface comprehensive performance tester and the coefficient of friction (COF) of the cladding coating is plotted in Figure 6. The COF curves were divided into two parts Ι and ΙΙ. The COF in the first minute (part Ι) varied greatly because the wear mechanism at this time primarily involved rupture of the oxide film on the surface. Therefore, the main wear mechanism of the cladding coating was represented in part ΙΙ, because the COF became stable at this time. All of the COF curves gradually decreased with the increase in the wear time, except the 0 HCT sample. Secondly, the longer the DCT duration, the smaller the COF value. This result was consistent with the microhardness tests [22,23,24]. Eventually, the lowest COF of all the samples was the 24 HCT sample, where the COF was reduced by about 30% compared with the 0 HCT sample. However, the COF instability of this sample increased due to the higher microhardness, seen in Figure 6b where the fluctuation of the 24 HCT curve was bigger than other samples. Therefore, it was more reasonable to perform 12 h of DCT to get the best COF for the cladding coating.

### 3.4. Worn Surface Morphology

As the surface of the material was always rough, the wear of the material was actually the interaction between the surface asperity [25]. The surface morphology after the wear test was the result of the interaction of the asperity under an external load. In this way, it reflected the wear resistance of the cladding coating from another perspective [26]. Therefore, Figure 7 shows the surface profile obtained by a non-contact optical surface profiler. It was distinct that the 0 HCT sample had a severe wear profile, which showed obvious internal deformation and large areas of collapse on both sides of the wear scar. This was because the 0 HCT sample had a low microhardness, so the asperity had severe deformation under the action of an external load and the surface was subjected to devastation [27,28]. However, the wear peak ridges of cladding coating with DCT only can be seen on both sides of the wear scar, and the ploughing phenomena of the internal wear scar were obvious (Figure 7b–f).

The profile curve fluctuation of the DCT sample gradually decreased as the microhardness increased (Figure 7h–l), the main wear mechanism after 9 h DCT was micro-ploughing because the surface asperity was more rigid and it was not easily broken under the external load. The wear resistance of the cladding coating was also influenced by the microstructure. The finer the grains of the cladding coating, and the more grains per unit volume, the more grains participated in resisting the deformation. As a result, the fine grains were not prone to microcracks in wear tests, thus improving the wear resistance of the surface [29,30,31]. This was why although the 24 HCT sample had the highest microhardness, the profile curve was similar to the 9 HCT and 12 HCT samples, because the 9 and 12 h samples had a better grain size and distribution. As a consequence, the finer microstructure could also enhance the wear resistance of the cladding coating.

### 3.5. Wear Factor

The wear depth, width and volume of wear scars are listed in Table 3.

According to the Hertz theory [31,32], combined with the Archard wear model [33], the wear coefficient of the cladding coating was calculated as follows:(1)$$V=K\frac{P\cdot L}{H}$$
where *V* is the wear volume; *P* is the direction pressure of the contact surfaces of the two workpieces; *L* is the tangential relative slip distance between the workpieces; *H* is the near-surface hardness; and *K* is the wear factor.

Thus, the expression of the wear factor *K* can be obtained by:(2)$$K=\frac{H\cdot V}{P\cdot L}$$

Archard pointed out that, the smaller the wear factor *K*, the better the wear resistance of the material [34]. The calculated wear factor results are shown in Figure 8. The DCT significantly enhanced the wear resistance of LC3530 Fe-based cladding coating; the wear resistance improved four times and five times, respectively, after 9 h and 12 h of DCT compared to the cladding coating without DCT. In addition, the 24 HCT sample had a similar wear factor to the 9 HCT sample, both were higher than the 12 HCT sample, which means that the best duration of DCT to improve the wear resistance for LC3530 Fe-based cladding coating was 12 h.

### 3.6. SEM Results of Wear Scars

The SEM results of wear scars could more clearly determine the wear mechanism of the cladding coating [35,36]. Figure 9 shows the main wear mechanism of all the samples under a magnification of 1000 times. The red box area shows the main wear characteristics that were used to conduct energy dispersive spectrometer (EDS) analysis, shown in Figure 10. In the morphology of the wear scar, the white area was a severely worn area, while the black area was the part with less wear. The cladding coating without DCT exhibited obvious extensive damage of the whole surface and a large amount of wear debris was generated, as shown in Figure 9a, which meant the surface was not wear resistant. However, the damage area of the 3 HCT sample decreased and the extrusion deformation (Figure 9b) could be seen under the same parameters. As for the 6 HCT sample (Figure 9c), it also had a smaller damage area and fewer debris because the surface microhardness increased, but fractures and micro-protrusion on the surface occurred because more heat was generated by the higher microhardness. Therefore, the 9 h, 12 h and 24 h samples had similar micro ploughing wear morphology because they all had higher microhardness. However, there was not a large area of damage in these sample’s surfaces and the main wear mechanism was micro-ploughing, as shown in Figure 9d–f. The morphology of the wear scars obviously proved that the DCT could enhance the wear resistance of the cladding coating. For the LC3530 Fe-based cladding coating, 9 h, 12 h and 24 h of DCT all exhibited better wear resistance.

The concentration of main elements was analyzed with EDS in a selected area of the wear characteristics as presented in the red box area of Figure 10, the wear characteristics were debris (Figure 10a), extrusion (Figure 10b), fracture (Figure 10c), micro-protrusion (Figure 10d), shedding (Figure 10e) and ploughing (Figure 10f). Firstly, the wear debris was many tiny white irregular objects. The surface deformation of the extrusion area was obvious and the particles produced by crushing could be seen. Secondly, it had a distinct lamellar structure in fracture areas and the crack was smooth. The shape of the micro-protrusion was spherical and drop-shaped. Finally, shedding and ploughing areas had significant small pieces falling off and a river-like trace, respectively.

It was obvious that the wear characteristics except the ploughing all had a large proportion of O atoms and a small proportion of Fe, C, and Cr atoms, while the ploughing was the opposite (Table 4). This indicated that the wear destroyed areas were composed of the oxide of Fe, which was not wear resistant. However, the proportion of Fe, C, and Cr atoms in the ploughing area was high because the cladding coating after DCT formed the wear resistant alloy compound composed of Fe, C, and Cr. Therefore, the DCT enhanced the wear resistance of the cladding coating by strengthening the surface microhardness through the precipitating carbides and alloy compound.

## 4. Conclusions

In this paper, the microstructure, microhardness and wear resistance of LC3530 Fe-based laser cladding coating after DCT were studied. The conclusions are as follows:

Deep cryogenic treatment refines the grain of the LC3530 Fe-based cladding coating through the dislocation generated by the internal stress due to the temperature difference effect, and by the precipitation of microcarbides during the cryogenic process, which improves the microhardness of the cladding coating. The microhardness after DCT was about 30% higher than the cladding coating without DCT.

According to the wear factor *K* of the Archard wear model, there is an improvement in wear resistance of four-fold and five-fold for 9 h and 12 h DCT cladding coating compared to the cladding coating without DCT under the same testing conditions. The 12 h hold time is optimal to get the lowest wear factor for the LC3530 Fe-based cladding coating.

Wear morphology analysis indicates that the cladding coating without DCT was severely worn due to the low microhardness; however, the cladding coating after DCT has smaller severely destroyed areas because of the higher microhardness. The main wear characteristics of 0 h, 3 h and 6 h samples were debris, extrusion and fracture, respectively, while the main wear mechanism of 9 h, 12 h and 24 h samples was typical micro-ploughing. The DCT enhances the wear resistance of cladding coating by forming the wear resistant alloy compound and promoting the precipitation of higher microhardness carbides.

## Figures and Tables

**Figure 1 materials-12-02400-f001:**
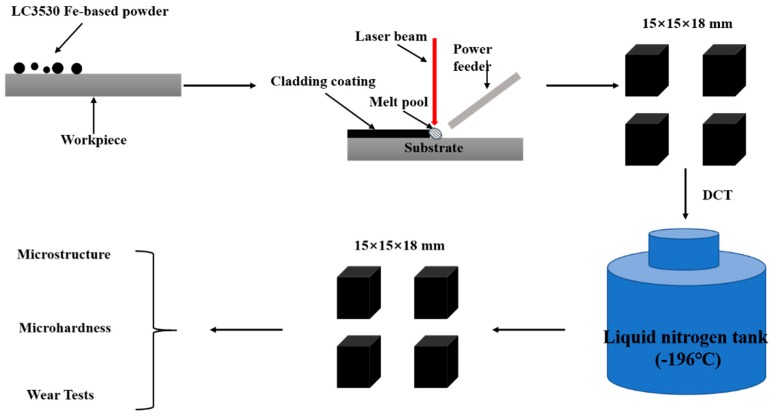
Experiment process profile.

**Figure 2 materials-12-02400-f002:**
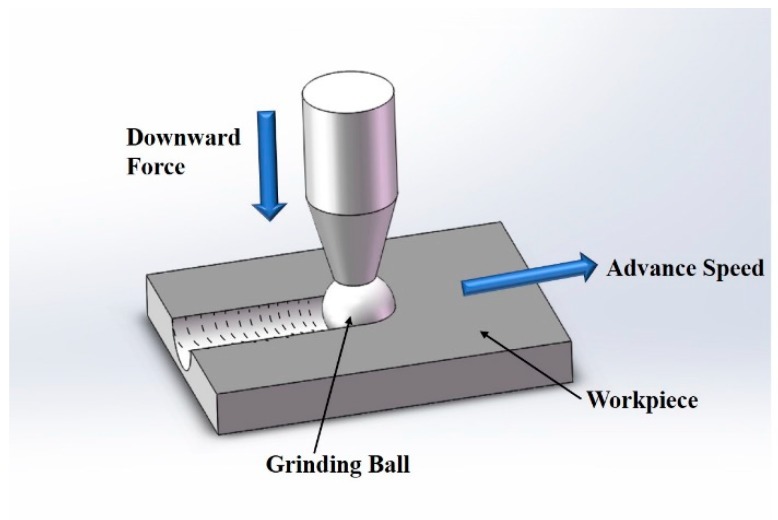
The schematic of the wear test.

**Figure 3 materials-12-02400-f003:**
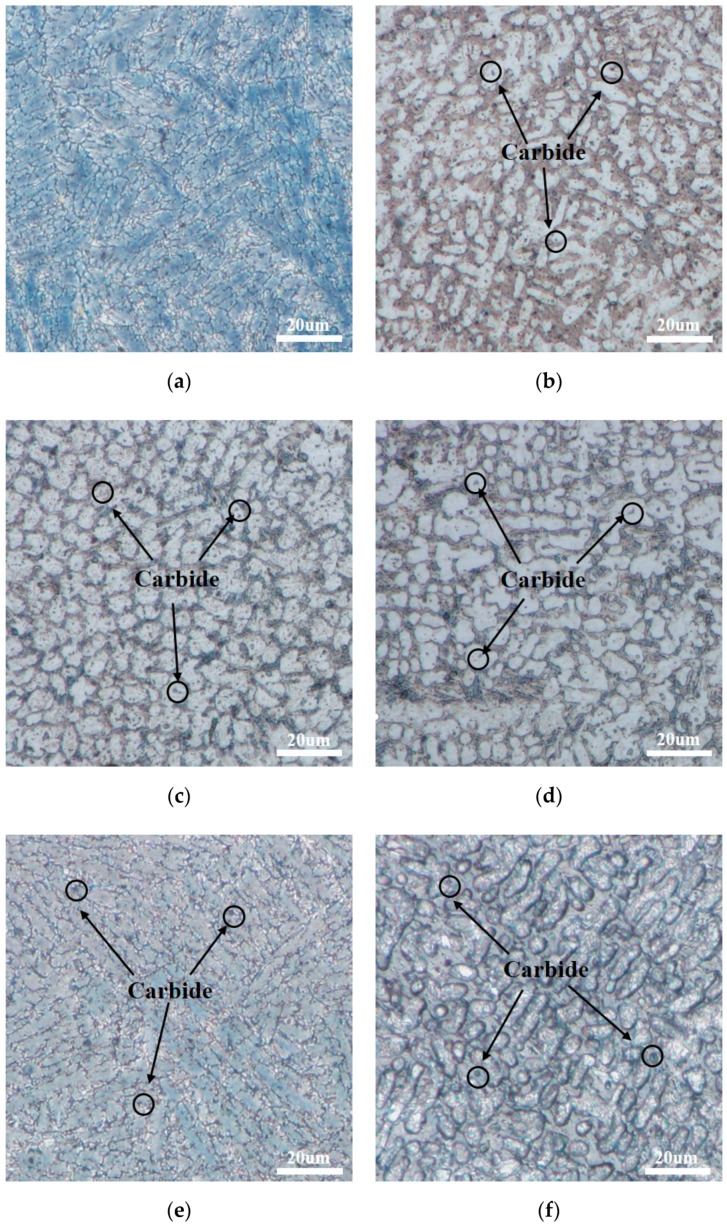
Microstructure of cladding coatings: (**a**) 0 HCT; (**b**) 3 HCT; (**c**) 6 HCT; (**d**) 9 HCT; (**e**) 12 HCT; (**f**) 24 HCT.

**Figure 4 materials-12-02400-f004:**
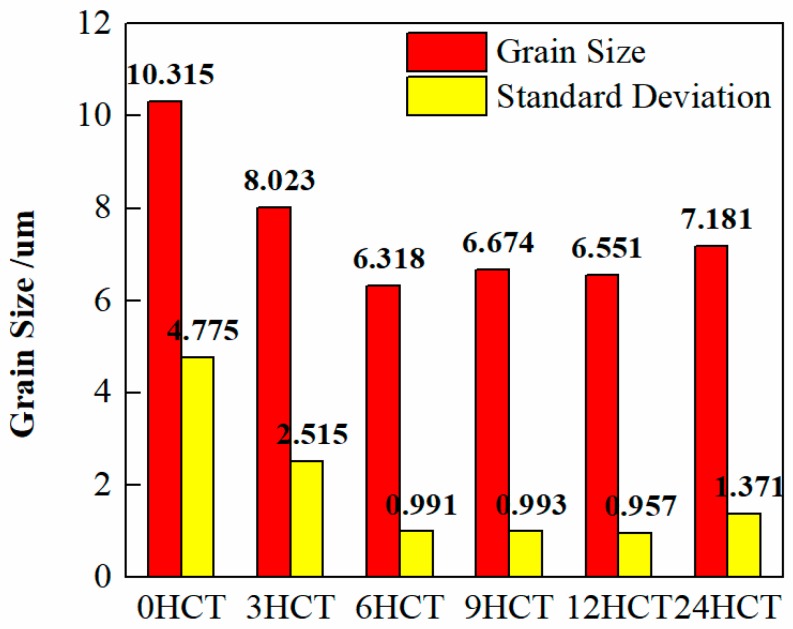
Average grain size and standard deviation of grain size.

**Figure 5 materials-12-02400-f005:**
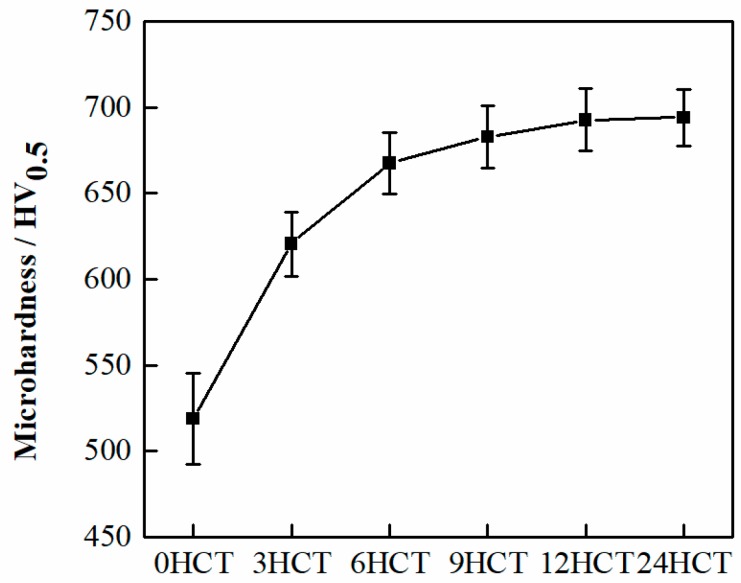
Average microhardness and standard deviation of microhardness.

**Figure 6 materials-12-02400-f006:**
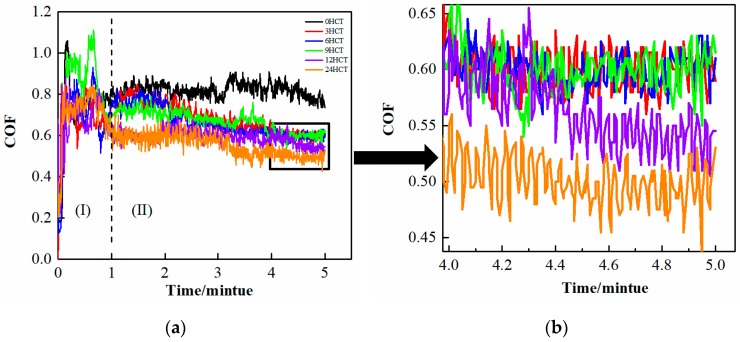
Coefficient of friction (COF) of the cladding coating in wear tests: (**a**) COF of the cladding coatings; (**b**) partial enlarged detail of COF curves.

**Figure 7 materials-12-02400-f007:**
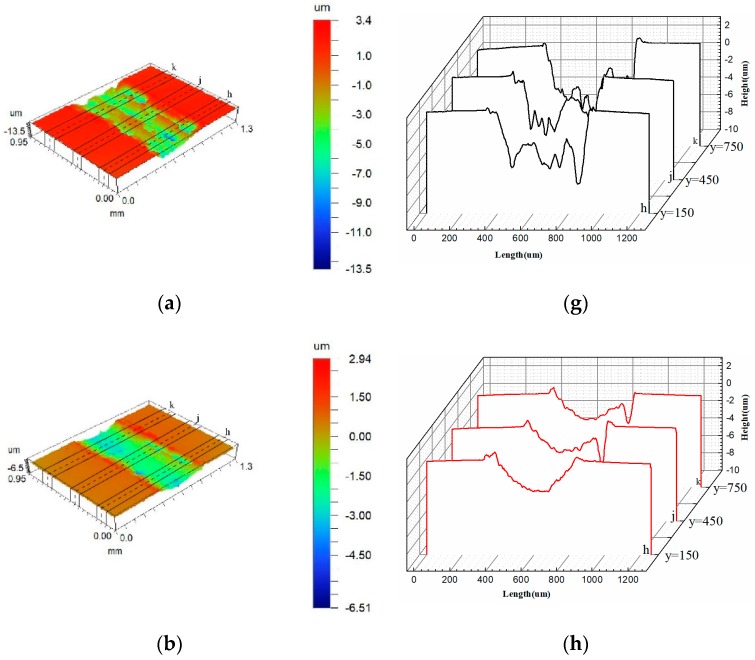
Three-dimensional surface profiles and profile curves of cladding coating after wear tests: (**a**) profile of 0 HCT; (**b**) profile of 3 HCT; (**c**) profile of 6 HCT; (**d**) profile of 9 HCT; (**e**) profile of 12 HCT; (**f**) profile of 24 HCT; (**g**) profile curve of 0 HCT; (**h**) profile curve of 3 HCT; (**i**) profile curve of 6 HCT; (**j**) profile curve of 9 HCT; (**k**) profile curve of 12 HCT; (**l**) profile curve of 24 HCT.

**Figure 8 materials-12-02400-f008:**
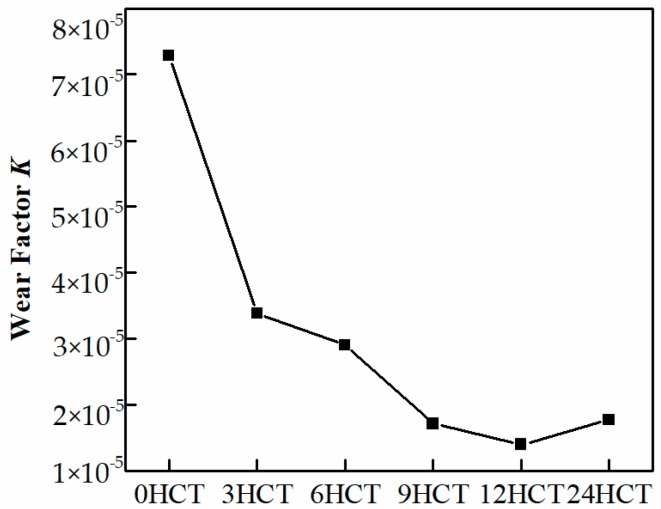
Wear factor of cladding coatings.

**Figure 9 materials-12-02400-f009:**
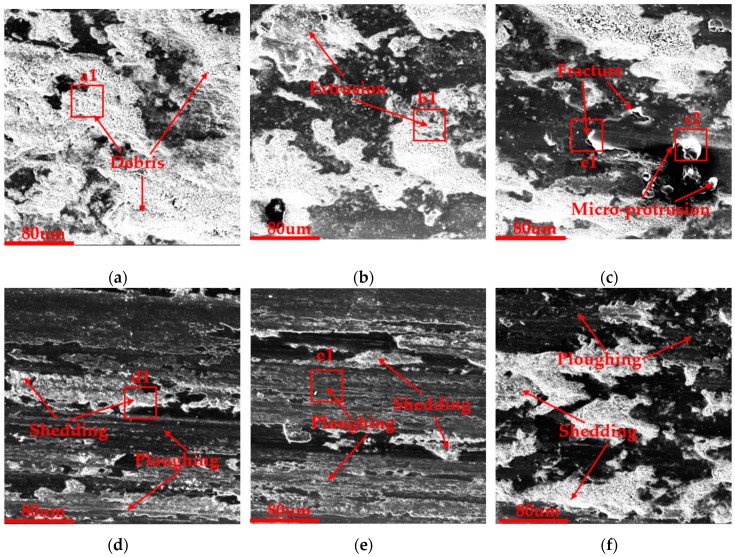
Wear scars morphology under SEM: (**a**) 0 HCT; (**b**) 3 HCT; (**c**) 6 HCT; and (**d**) 9 HCT; (**e**) 12 HCT; (**f**) 24 HCT.

**Figure 10 materials-12-02400-f010:**
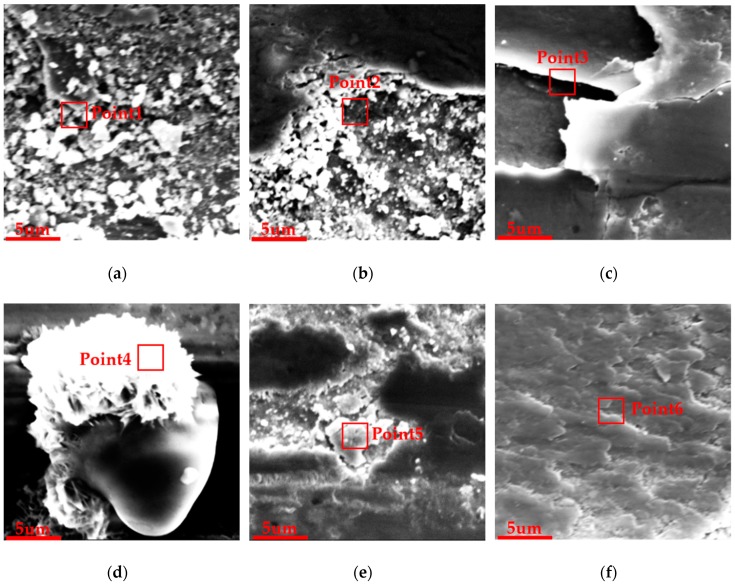
SEM results of the wear characteristics: (**a**) debris; (**b**) extrusion; (**c**) fracture; (**d**) micro-protrusion; (**e**) shedding; (**f**) ploughing.

**Table 1 materials-12-02400-t001:** Chemical composition of 35CrMo steel (wt%).

C	Si	Mn	Cr	Mo	Ni	P	S	Fe
0.36	0.30	0.61	0.98	0.19	0.012	0.014	0.0062	Bal

**Table 2 materials-12-02400-t002:** Chemical composition of laser cladding material LC3530 (wt%).

C + N	Cr + Ni	Nb + Ta	Fe	Other
0.07–0.50	18.1–25.0	Max 1.3	Base	2.5–4.5

**Table 3 materials-12-02400-t003:** Wear depth, width and volume of wear scars.

Sample	0 CHT	3 HCT	6 HCT	9 HCT	12 HCT	24 HCT
Wear Depth/um	2.96	1.37	1.17	0.69	0.57	0.71
Wear Width/um	523.5	539.4	493.8	493.8	489.8	452.1
Wear Volume/um^3^	1.40 × 10^6^	6.5 × 10^5^	5.6 × 10^5^	3.3 × 10^5^	2.7 × 10^5^	3.4 × 10^5^

**Table 4 materials-12-02400-t004:** EDS results of the wear characteristics (Atomic Percentage).

Test point	O	C	Fe	Cr	Mo	Ni
Point 1	49.54	30.43	19.22	0.73	0.08	0.00
Point 2	39.84	18.84	37.11	3.95	0.19	0.08
Point 3	53.64	24.48	20.27	1.45	0.09	0.07
Point 4	49.28	14.01	35.09	1.12	0.28	0.22
Point 5	47.24	27.87	23.30	1.43	0.11	0.05
Point 6	17.23	42.06	32.88	6.97	0.46	0.39

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
