# Peer review of "Effect of Deep Cryogenic Treatment on Microstructure and Wear Resistance of LC3530 Fe-Based Laser Cladding Coating"

_materials, 2019, doi:10.3390/ma12152400_

Round 1
Reviewer 1 Report
The paper describes the effect of cryogenic treatment on laser cladded coatings. The idea is interesting, and the results indicate a positive influence of cryogenic treatment on wear resistance. However, the paper is lacking in several important areas:
The Introduction needs to be improved significantly. It must explain the process of laser cladding in greater detail by citing work done in the past.
More details are needed regarding the powder. What is the size distribution? Supplier? Why did you choose this particular alloy?
How did you arrive at the laser processing parameters? Are they optimized?
Micrographs of the entire cladded coating are needed to indicate the thickness of the coating. The labels (a, b, c etc) are also not right in Fig 3. Moreover, how were the carbide particles (labeled) identified? Are they not present in the 0HCT specimen?
How was the grain size measured in Fig 4?
How are internal stresses developed if the cooling is uniform over a period of hours?
The wear depth in Table 3 seems too low. How was it calculated? The wear profiles look pretty rough in Fig 7. If it is an average value, there should also be a standard deviation.
H in formula 2 (Archard's equation) is the near-surface hardness and not the highest hardness.
SEM image quality in Figs 9 and 10 is not good enough.
Author Response
We are truly grateful to yours and other reviewers’ critical comments and thoughtful suggestions. Based on these comments and suggestions, we have made careful modifications on the original manuscript. All changes made to the text are in blue colour. In addition, we have consulted native English speakers for paper revision before the submission this time. We hope the new manuscript will meet your magazine’s standard. Below you will find our point-by-point responses:
Point 1: The Introduction needs to be improved significantly. It must explain the process of laser cladding in greater detail by citing work done in the past.
Response 1: Thanks for your advice. We are sorry about that and the laser cladding process, process characteristics and past work results have been added to the introduction. (Line 31-47)
Point 2: More details are needed regarding the powder. What is the size distribution? Supplier? Why did you choose this particular alloy?
Response 2: We are sorry that many details are not mentioned. The particle size distribution was 45-180μm and was supplied by Xiangtan City Unicom Industry and Trade Co., Ltd. The reason to chooses this alloy was that it could form a good metallurgical bond with the substrate. And these have been added in this paper. (Line 80-82)
Point 3: How did you arrive at the laser processing parameters? Are they optimized?
Response 3: Yes, in our previous studies we have made different tests to obtain the optimal laser cladding parameters.
Point 4: Micrographs of the entire cladded coating are needed to indicate the thickness of the coating. The labels (a, b, c etc) are also not right in Fig 3. Moreover, how were the carbide particles (labeled) identified? Are they not present in the 0HCT specimen?
Response 4: The thickness of the coating is about 3mm and the labels are corrected. The metallographic examinations include microstructure analysis, grain size, and carbide determination, where the determination of carbides is determined by the morphology of the carbide. We haven’t found the carbide particles in the 0HCT specimen.
Point 5: How was the grain size measured in Fig 4?
Response 5: The grain size was measured by ImageJ software.
Point 6: How are internal stresses developed if the cooling is uniform over a period of hours?
Response 6: The internal stress is caused by thermal expansion and contraction, and thermal expansion and contraction is the result of atomic thermal motion, so if the cooling is uniform over a period of hours, the atomic thermal motion does not change and neither does the inter stress.
Point 7: The wear depth in Table 3 seems too low. How was it calculated? The wear profiles look pretty rough in Fig 7. If it is an average value, there should also be a standard deviation.
Response 7: The wear depth was calculated in wear profiles by the Vision software. The wear profiles look rough because the value of the coordinate axis is small, it is the data value obtained directly from the wear profile.
Point 8: H in formula 2 (Archard's equation) is the near-surface hardness and not the highest hardness.
Response 8: Thank you for your correction, we have made the changes in the paper. (Line 246)
Point 9: SEM image quality in Figs 9 and 10 is not good enough.
Response 9: Thank you for pointing out that and the images have been replaced. (Line 277, 278, and Line 299, 300)
Please see the attachment.

Reviewer 2 Report
This work investigates the effect of deep cryogenic treatment on the microstructure and wear resistance of LC3530 Fe-based laser cladding coating. The research is interesting, however, to be accepted for publication the following comments are required to be addressed:
1-The Introduction needs to be further improved. Please explain about laser cladding process and its benefit and the reasons for improving the mechanical properties before starting to explain about previous research carried out from line 3 of the introduction section.
2-The English of the manuscript needs to be further improved. For example, line 51 should be revised to “had not been reported yet.” Line 52 should be changed to “needed to be improved …”, etc.
3- It is well known that the phases present in the microstructure affect the hardness of the materials which in turn affect the wear resistance of the materials. Please read and use the following articles: Materials & Design 111 (2016), 592-599, Materials Science and Engineering: A 760 (2019), 339-345 and Journal of Alloys and Compounds 787 (2019), 570-577.
4- In Fig. 5, please explain the reason for small increase in microhardness as the DCT duration time increases.
Author Response
We are truly grateful to yours and other reviewers’ critical comments and thoughtful suggestions. Based on these comments and suggestions, we have made careful modifications on the original manuscript. All changes made to the text are in blue colour. In addition, we have consulted native English speakers for paper revision before the submission this time. We hope the new manuscript will meet your magazine’s standard. Below you will find our point-by-point responses:
Point 1: 1-The Introduction needs to be further improved. Please explain about laser cladding process and its benefit and the reasons for improving the mechanical properties before starting to explain about previous research carried out from line 3 of the introduction section.
Response 1: Thanks for your advice. We are sorry about that and the laser cladding process, process characteristics and past work results have been added to the introduction. (Line 31-47)
Point 2: 2-The English of the manuscript needs to be further improved. For example, line 51 should be revised to “had not been reported yet.” Line 52 should be changed to “needed to be improved …”, etc.
Response 2: Thank you for pointing out that and we have consulted native English speakers for paper revision.
Point 3: 3- It is well known that the phases present in the microstructure affect the hardness of the materials which in turn affect the wear resistance of the materials. Please read and use the following articles: Materials & Design 111 (2016), 592-599, Materials Science and Engineering: A 760 (2019), 339-345 and Journal of Alloys and Compounds 787 (2019), 570-577.
Response 3: These are great articles and thanks for your recommendation, we have read and use these articles as our references.
Point 4: 4- In Fig. 5, please explain the reason for small increase in microhardness as the DCT duration time increases.
Response 4: This is because the microhardness is determined by the microstructure and carbides. The change of grain size and carbides precipitation after 6 hours DCT is small, so the increment of microhardness value is slow. It has been supplemented in the paper. (Line 177-179)
Please see the attachment.

Reviewer 3 Report
The abstract must contain the objetive and the context of the study.
The introduction must be highly improved. It's necessary incorporate a lot of information about laser cladding. Only it is described others investigations and their results, but it is necessary establishing the relationship with the general context, their applications and the actual research. I recommend incorporate the investigation for example of A. Lamikiz from University of Basque Country.
And recommend revise the general format of the paper.
I recommend incorporate the norms used in experimental procedure and justify the materials, parameters used and all the techniques described.
It's necessary analyze the surface with EDS or other technique that garantize the precipitation observed.
How was measure the microstructure? It is necessary incorporate this information.
It is necessary improve the discussion of the results.
Where are the microhardness measure localized? Why?
How can be explain the dispersión of microhardness?
The general appearance of the images must be improved.
A general model of the behaviour of the friction test must be included.
The Figure 7 appear in 4 pages. It is very unusual and complicate the visualization.
The wear factor must be discussed previously.
The EDS localization must appear. Figures 9 y 10 must be improved.
The authors don't use the template.
Author Response
We are truly grateful to yours and other reviewers’ critical comments and thoughtful suggestions. Based on these comments and suggestions, we have made careful modifications on the original manuscript. All changes made to the text are in blue colour. In addition, we have consulted native English speakers for paper revision before the submission this time. We hope the new manuscript will meet your magazine’s standard. Below you will find our point-by-point responses:
Point 1: The abstract must contain the objective and the context of the study.
Response 1: Thanks for pointing that and we have modified the abstract according to your suggestions.
Point 2: The introduction must be highly improved. It's necessary incorporate a lot of information about laser cladding. Only it is described others investigations and their results, but it is necessary establishing the relationship with the general context, their applications and the actual research. I recommend incorporate the investigation for example of A. Lamikiz from University of Basque Country.
Response 2: We are sorry about the introduction and we have added a lot of information about laser cladding and pointed out the connection between laser cladding and industrial applications. And thanks for your recommendation, we have read and use the article of A. Lamikiz as our references. (Line 31-47)
Point 3: And recommend revise the general format of the paper.
Response 3: Thank you for pointing that and we have made the correction.
Point 4: I recommend incorporate the norms used in experimental procedure and justify the materials, parameters used and all the techniques described.
Response 4: We have supplemented the specification of the experimental process, supplementing the particle size of the alloy powder, the supplier and the reason for the selection. (Line 80-82)
Point 5: It's necessary analyze the surface with EDS or other technique that garantize the precipitation observed.
Response 5: Yes, the precipitation observed was analysed by EDS, SEM and the result of the microstructure.
Point 6: How was measure the microstructure? It is necessary incorporate this information.
Response 6: Yes, it is necessary to explain how the microstructure was measured. We corrode and observe the sample according to the metallographic method, and then measure the microstructure according to the metallographic standard. The paper has been supplemented. (Line 106)
Point 7: It is necessary improve the discussion of the results.
Response 7: Thank for your advice, we have improved the discussion of the results based on the experimental results.
Point 8: Where are the microhardness measure localized? Why?
Response 8: The microhardness measurement position is located in the center of the surface, a total of two lines, 5 test points per line, and we have already added in the paper. Because the center has the best cladding effect, it could reduce the impact of the laser cladding process on the measurement. (Line 111-114)
Point 9: How can be explain the dispersión of microhardness?
Response 9: The dispersion of microhardness is caused by the difference in the microstructure and carbide content at different positions. If the microstructure at this position is uniform and the carbide content is large, the microhardness is high. (Line 173-176)
Point 10: The general appearance of the images must be improved.
Response 10: Yes, we have improved the appearance of the images.
Point 11: A general model of the behaviour of the friction test must be included.
Response 11: Yes, a general model of the behaviour of the friction test is necessary, so we use the Figure 2 as the model for the friction test.
Point 12: The Figure 7 appear in 4 pages. It is very unusual and complicate the visualization.
Response 12: Figure 7 is a three-dimensional profile and a two-dimensional profile curve after wear tests, which can more intuitively reflect the influence of the deep cryogenic treatment on the wear resistance of the cladding coating.
Point 13: The wear factor must be discussed previously.
Response 13: Yes, the wear factor should be discussed previously. However, it is calculated by the wear volume, and the wear volume is calculated by the wear depth which is obtained through the wear profiles. Based on this, we put the wear factor discussion after the wear profiles.
Point 14: The EDS localization must appear. Figures 9 y 10 must be improved.
Response 14: Yes, the EDS tests a total of six localizations, which are the red areas in Figures 10. And the Figures 9 and 10 has been replaced. (Line 277, 278, and Line 299, 300)
Point 15: The authors don't use the template.
Response 15: Thank you for pointing that, we have modified this paper according to the template.
Please see the attachment.

Round 2
Reviewer 1 Report
The idea of cryogenic treatment is good, but the paper is still lacking in many key areas, especially the Introduction.
Author Response
Dear Reviewer:
RE: materials-539019
Thank for your careful read and thoughtful comments on our paper. We have carefully taken your comments into consideration in preparing our revision. The following is the list of changes.
Point 1: The idea of cryogenic treatment is good, but the paper is still lacking in many key areas, especially the Introduction.
Response 1: Thank you for your great advice. Introduction sections of the articles are like gates of a city. It is a presentation aiming at introducing itself to the readers, and attracting their attention. Therefore, we have added information about why we choose the substrate and powder of the manuscript. And the problem during the laser cladding progress related to the topic. Based on this, the DCT is our recommendations for solution. We have already supplemented the introduction. (Line 68-80)
As for the language of this paper, we have made the English editing by MDPI. Thanks for your suggestions.
Please see the attachment.

Reviewer 2 Report
The authors have revised the manuscript according to the comments and the quality of the work has been improved. Therefore, the manuscript can be accepted for publication.
Author Response
Dear Reviewer 2:
Special thanks to you for your great comments.
Reviewer 3 Report
The authors have follow the recommendations. The paper has been improved.
Author Response
Dear reviewer 3:
Special thanks to you for your great comments.